# Mapping the role of tumor necrosis factor-related apoptosis-inducing ligand (TRAIL) and its receptors in chronic kidney disease: A scoping review protocol

Hansani Niroshika[iD][1,2]*, Harshi Weerakoon[3], Nalaka Herath[4], Kosala Weerakoon[5]*, Prasad Jayasooriya[2]

**1** Faculty of Graduate Studies, General Sir John Kotelawala Defence University, Ratmalana, Sri Lanka, **2** Department of Bioprocess Technology, Faculty of Technology, Rajarata University of Sri Lanka, Mihintale, Sri Lanka, **3** Department of Biochemistry, Faculty of Medicine and Allied Sciences, Rajarata University of Sri Lanka, Saliyapura, Sri Lanka, **4** Colombo North Teaching Hospital, Ragama, Sri Lanka, **5** Department of Parasitology, Faculty of Medicine and Allied Sciences, Rajarata University of Sri Lanka, Saliyapura, Sri Lanka

\* hansaniniroshika014@gmail.com (HN); kosalagadw83@gmail.com (KW)

## Abstract

Chronic kidney disease (CKD) is increasingly recognized as a global public health concern, impacting about 800 million people worldwide. Despite having different CKD etiologies, apoptosis, inflammation and fibrosis are the central mechanisms of pathogenesis. Tumor necrosis factor (TNF)- related apoptosis-inducing ligand (TRAIL) is a critical regulator of these mechanisms, suggesting a prominent role in pathogenesis and progression of CKD. Therefore, systematic mapping of the available evidence is pivotal in identifying the exact role of TRAIL and its receptors in CKD. The proposed scoping review will be conducted in line with Arksey and O'Malley's methodological framework and the Joanna Briggs Institute (JBI) reviewer's manual. Accordingly, the review will be guided by the five-stage approach, namely (1) identify the research question; (2) identify relevant studies; (3) select studies; (4) chart the data; and (5) collate, summarize, and report the results. Eligibility criteria and search strategies will be formulated based on population, concept, and context (PCC) strategy. Articles published up to January 2026 will be searched using electronic databases (PubMed/ MEDLINE, Science Direct, Scopus, CINAHL, EMBASE, Web of Science and Cochrane Library) and clinical trial registries (ClinicalTrials.gov and WHO ICTRP). Reference lists of relevant reviews retrieved from Google Scholar will also be screened for grey literature. A formal quality appraisal of the selected studies will be conducted using the mixed-method appraisal tool (MMAT version 2018). The reporting will be done following the Preferred Reporting Items for Systematic Reviews and Meta-Analyses extension for the Scoping Reviews checklist. The literature search is expected to commence in March 2026, followed by study selection and data extraction by April 2026. Data synthesis will be completed by June 2026. The

**Data availability statement:** No datasets were generated or analysed during this scoping review protocol. All information required for the protocol is included in the paper and its Supporting Information materials. Data generated during the conduct of the scoping review will be made publicly available upon completion of the study.

**Funding:** The authors acknowledge that this scoping review protocol is part of a major project under science and technology human resource development project, Ministry of Higher Education, Sri Lanka, funded by the Asian Development Bank (Grant No. R2RJ4, under Prof. Prasad Jayasooriya). However, no specific funding was received for this protocol.

**Competing interests:** The authors declare that they have no known competing financial interests or personal relationships that could have appeared to influence the work reported in this paper.

review protocol is registered at Open Science Framework (OSF) under https://doi.org/10.17605/OSF.IO/AENBX.

## Introduction

Chronic kidney disease (CKD) is defined as "abnormalities of kidney structure or function, present for a minimum of 3 months, with implications for health" [1] is a global health burden affecting more than 800 million people worldwide [2]. Hypertension and diabetes mellitus are the most common causes of CKD, while factors like genetics, infections, adverse drug effects, and autoimmunity can also cause the disease [3,4]. Regardless of the underlying cause, CKD increases the risk of premature deaths caused by cardiovascular diseases [5]. On a histological basis, the disease is characterized by the gradual depletion of different cell types including podocytes, tubular epithelial cells, and endothelial cells resulting in glomerulosclerosis, tubular atrophy, and capillary rarefaction. Simultaneously, maladaptive cells like myofibroblasts and inflammatory cells are recruited and proliferated, causing a significant loss of renal function due to fibrosis [6,7].

Tumor necrosis factor (TNF)- related apoptosis-inducing ligand (TRAIL) is a TNF superfamily cytokine (member 10) that is highly likely to be involved in CKD pathology. TRAIL can induce apoptosis selectively in transformed cells with minimal off-target effects [8,9]. This selectivity arises from the differential expression of death (TRAIL-R1/DR4 and TRAIL-R2/DR5), decoy (TRAIL-R3/DcR1, TRAIL-R4/DcR2), and soluble osteoprotegerin (OPG) receptors of TRAIL. Death receptors, when activated, trigger apoptotic signaling while decoy and OPG receptors promote cell survival and evade TRAIL-induced apoptosis [10]. While its role has been studied extensively as an apoptotic inducer of cancer cells, its application in noncancerous diseases, particularly fibrotic and inflammatory conditions such as CKD, is an emerging area of interest.

A recent study reported an inverse relationship between TRAIL levels and mortality risk in CKD [11]. In addition, a significant association between low levels of circulating TRAIL and atheromatous plaque progression has been identified, and these results highlighted the possible use of TRAIL as an independent prognostic biomarker to monitor atherosclerosis in patients with CKD [12]. Further, some recent experimental animal studies have emphasized the potential role of TRAIL in CKD. For example, an experiment conducted using TRAIL(-/-) ApoE(-/-) and ApoE(-/-) mice fed with a high-fat diet for 20 weeks revealed exacerbated diabetic nephropathy in the presence of TRAIL deficiency [13]. On the other hand, TRAIL treatment (twice daily over 12 weeks) seems to remodel the glomerular and tubular morphology, causing improved renal functions in db/db (diabetic) mice [14].

Mechanistically, this ligand appears to influence tubulointerstitial injury, glomerular filtration, and vascular remodeling based on the receptor expression modulated by nature of cytotoxic insult and microenvironment. The balance between proapoptotic and antiapoptotic TRAIL receptors determines whether TRAIL signaling promotes cell death, survival, inflammation, or senescence of renal cells. For example,

in early diabetic kidney disease (DKD) TRAIL has been associated with podocyte PANoptosis by expressing higher DR5 levels [15]. Concurrently, metabolic stress and advanced glycation end products are shown to promote tubular cell senescence and maladaptive repair by expressing higher DcR2 levels [16,17]. Likewise, the precise role of TRAIL and its receptors in CKD of different etiologies remains under investigation and literature remains fragmented with limited synthesis of data.

Therefore, mapping the existing evidence systematically will pave the way to identify the significance of TRAIL in CKD and thereby identify the areas that need to be further explored. For this, we selected scoping review approach since existing studies span across different experimental models, clinical observations, and molecular mechanisms among different CKD subtypes. The outcome review paper aims to summarize and highlight the existing research gaps related to TRAIL/TRAIL-R expression patterns, regulatory mechanisms and associations with clinical outcomes across different CKD etiologies. This scoping review protocol will define the transparent and systematic methodology to identify, chart and synthesize existing evidence.

## Methodology

This review will comply with the methodological framework described by Arksey and O'Malley [18] and revisions made by the Joanna Briggs Institute (JBI) [19]. Thus, the review will be developed in five stages (Fig 1). In phase 3, reference lists of all identified papers, relevant literature reviews, and the Google Scholar database will be used to find additional and grey literature.

Scoping review reporting will adhere to the Preferred Reporting Items for Systematic Reviews and Meta-Analyses extension for Scoping Reviews (PRISMA-ScR) checklist (S2 Table in S2 File) [20]. The review will be conducted from March 2025 to June 2026, following a structured timeline. The database search is planned to be initiated on 1st of March 2026 including all articles up to January 2026. Study selection and data extraction will be completed by April 2026, followed by data synthesis up to June 2026. The manuscript will be drafted and finalized in July 2026. As this study consolidates secondary data, ethical approval will not be required.

### Stage 1: Identification of research question

The population, concept, and context (PCC) framework [21] is used to formulate the primary review question and to develop the search strategy (Table 1).

The main research question of this review is 'What are the patterns, mechanisms, and clinical implications of TRAIL and its receptor expression in CKD?'

Areas explored under the above overarching question are,

1. What are the patterns of TRAIL and its receptor expression in CKD?

2. What are the mechanisms underlying the regulation of TRAIL expression in CKD?

3. What are the associations of TRAIL with disease severity, progression, and clinical outcomes in CKD?

4. What are the knowledge gaps in understanding the role of TRAIL across CKDs of different etiologies?

### Stage 2: Database search and identification of relevant studies

The article search will follow the JBI-recommended three-phase search process [21]. In phase 1, an initial search will be conducted through PubMed with two main search terms and their related key words (i) Tumor necrosis factor-related apoptosis-inducing ligand and (ii) chronic kidney disease. Then, a second search (phase 2) will be performed with the identified most relevant keywords in phase 1. For this, database-specific search strategies will be implemented on electronic databases (PubMed/ MEDLINE, Science Direct, Scopus, CINAHL, EMBASE, Web of Science, and Cochrane

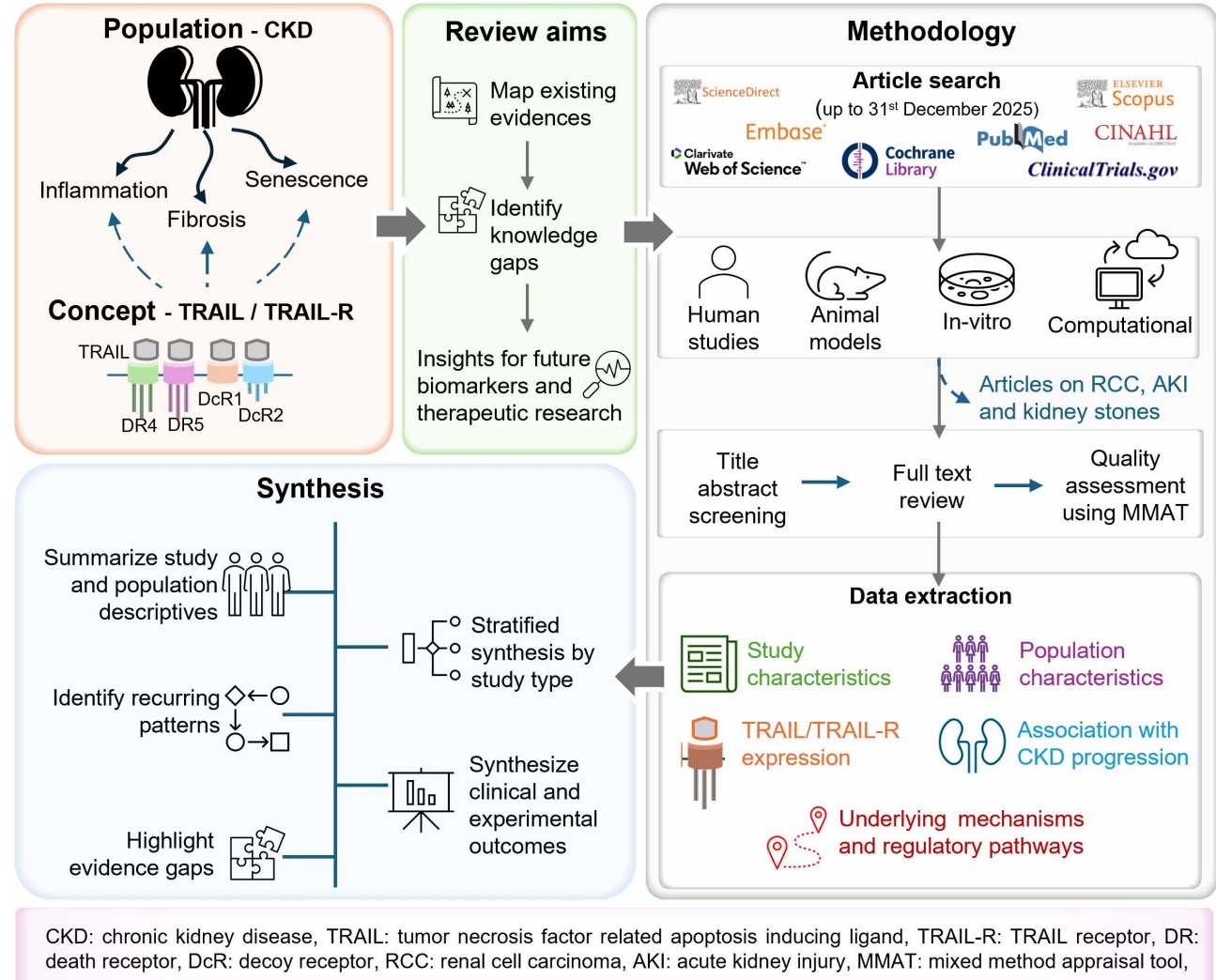

**Fig 1. Conceptual framework of the proposed scoping review.**

**Table 1. PCC framework for the proposed scoping review.**

| Criteria | Determinants |
|---|---|
| P- Population | All human studies, animal models, cell lines, and related computational models of CKD |
| C- Concept | TRAIL/ TRAIL receptors in CKD research |
| C- Context | Not limited to a specific region or country |

Library) and clinical trial registries (ClinicalTrials.gov and WHO ICTRP). All peer-reviewed articles published up to January 2026 will be included (Article search will be conducted on 1st March 2026 and full Boolean search string for each database is included in S1 table in S1 File).

## Stage 3: Selection of eligible studies

Article selection will follow the PCC framework along with the following eligibility criteria to confirm the most relevant studies to review questions.

### Inclusion criteria.

1. Quantitative and qualitative human/animal studies or computational models on expression, regulation, mechanisms, and clinical implications of TRAIL in CKD

2. Articles published up to January 2026

3. Reviews in line with the search string will be considered for background details and to trace relevant articles that were not detected in the initial screening process

### Exclusion criteria.

1. Articles on renal carcinoma, acute kidney injury, and kidney stones

Depending on the above criteria, a two-step approach will be carried out to filter the articles.

**Step 1: Title and abstract screening.** The articles retrieved from the database search will be imported into reference management software for deduplication. Two independent reviewers will screen the titles and abstracts and assess whether they meet the inclusion criteria. In case of any disagreement, the opinion of a third reviewer will be obtained.

**Step 2: Full text review.** Selected studies from Step 1 will be subjected to comprehensive full-text screening to identify their suitability for subsequent data extraction. In selecting and mapping included studies, recommendations from the PRISMA-ScR checklist and PRISMA-P chart will be followed [20]. The Rayaan.ai AI-powered systematic review management platform will be used to store and manage data throughout the review process [22].

**Step 3: Critical appraisal.** The mixed-method appraisal tool (MMAT version 2018) will be used to assess the quality of the selected studies. This tool comprises assessment criteria for qualitative, quantitative randomized control trials (RCTs), quantitative non-RCTs, quantitative descriptive, and mixed methods. Each item will be rated on a scale of "yes", "no", or "can't tell" and each study will receive an overall score based on the number of criteria met divided by 5, resulting in scores from 20% (one criterion met) to 100% (all criteria met) [23]. The appraisal will be initially conducted by one reviewer and independently assessed by a second reviewer. Any discrepancy will be discussed and resolved by consensus. Scores will be used as a guide to interpret study quality and report descriptively, but no studies will be excluded solely based on low scores.

## Stage 4: Charting the data

Articles selected from the above stage will be used for data extraction. Data will be charted using the data extraction framework (Table 2) that captures bibliographic details, study design, population characteristics, TRAIL related biomarkers, and outcomes in CKD progression. This will be conducted under 5 key subject areas, and this preliminary content may be amended as the review progresses.

1. Study characteristics

2. Population characteristics

3. TRAIL expression

4. Underlying mechanisms and regulatory pathways

5. Association with CKD progression

**Table 2.  Data extraction framework.**

| No. | Main Category | Subcategory | Description |
|---|---|---|---|
| Study characteristics | | | |
| 1 | Authors | | |
| 2 | Title | | |
| 3 | Year of publication | | |
| 4 | Country of origin | Geographical area | |
| 5 | Objectives | Main objective | Specify the main objective |
| | | Specific objectives | Specify the specific objectives |
| 6 | Study design | | Specify the study approach (e.g., Observational, experimental, cohort, case-control, cross-sectional) |
| 7 | Study setting | | Specify the study environment (e.g., Community-based, hospital-based, or laboratory-based) |
| Population characteristics | | | |
| 8 | Type of CKD etiology | | Specify the etiology of CKD studied (e.g., diabetic nephropathy, hypertensive nephropathy, CKDu) |
| 9 | CKD stage | | Specify CKD stages considered (stages 1–5) |
| 10 | Experimental model (if applicable) | Human studies | Specify the age range and male-to-female ratio |
| | | Animal studies | Specify the model (e.g., mice, zebrafish) |
| | | Experimental studies | Specify the model (e.g., cell lines) |
| | | Computational studies | Specify the model (e.g., in silico simulations, molecular modeling) |
| 11 | Sample size | Cases and controls | Specify the participant count, if cell culture specify the replicates |
| 12 | Comorbidities (Human studies) | | Specify the comorbidities |
| 13 | Method of inducing CKD (Animal studies) | | Specify the method of inducing CKD |
| TRAIL expression | | | |
| 14 | Sample of analysis | | Specify the type of sample (e.g., tissue, blood, urine) |
| 15 | Type of analyte | TRAIL | Specify the type of TRAIL (e.g., soluble, membrane-bound, intracellular) |
| | | TRAIL receptor | Specify the type of TRAIL receptor (DR4, DR5, DcR1, DcR2, OPG) |
| 16 | Method of analysis | TRAIL | Specify the method (e.g., ELISA, flowcytometry, western blot) and kits used |
| | | TRAIL receptor | Specify the method (e.g., ELISA, flowcytometry, western blot) and kits used |
| 17 | Expression level | CKD sample | Specify the TRAIL or its receptor expression level |
| | | Control sample | |
| Mechanisms and regulatory pathways | | | |
| 18 | Pathways and mechanisms | | Specify the associated signaling pathways |
| 19 | Influencing factors | | Specify the factors that influence TRAIL expression (e.g.,: inflammation, environmental exposures) |
| 20 | Molecular markers | | Specify any additional markers studied |
| Association with disease progression | | | |
| 21 | Disease stages | | Specify the association of TRAIL with the stage of the disease |
| 22 | Progression markers | | Specify the relation of TRAIL with any progression marker (e.g., eGFR, UACR) |

*(Continued)*

**Table 2.** (Continued)

| No. | Main Category | Subcategory | Description |
|---|---|---|---|
| 23 | Clinical outcomes | | Specify the association of TRAIL with clinical outcomes (e.g., mortality, dialysis initiation, cardiovascular events) |
| 24 | Conclusions | | Specify the conclusions of the study |
| 25 | Limitations | | Specify the limitations of the study |

## Stage 5: Collating and summarizing the results

The outcome of this scoping review will address the research questions mentioned above (section: stage 1). PRISMA flow diagram will be used to illustrate the article selection process including the total number retrieved, duplicates removed, and the final number included.

Data from heterogenous populations (human studies, animal models, cell lines, and related computational models) will be synthesized separately. Consistent with the objectives of this scoping review, a meta-analysis will not be conducted. After contextualizing based on research questions, results will be presented as narrative descriptions while including counts, frequencies, and proportions arise from quantitative data as appropriate. Low quality studies (MMAT scores <50%) will be flagged and discussed cautiously to avoid misinterpretation of potentially biased findings.

## Supporting information

**S1 File. Full Boolean search string for each database.**
(DOCX)

**S2 File. Preferred Reporting Items for Systematic Reviews and Meta-Analyses protocol (PRISMA-P) checklist.**
(DOCX)

## Author contributions

**Conceptualization:** Hansani Niroshika, Kosala Weerakoon.

**Methodology:** Hansani Niroshika, Kosala Weerakoon.

**Supervision:** Harshi Weerakoon, Nalaka Herath, Kosala Weerakoon, Prasad Jayasooriya.

**Visualization:** Hansani Niroshika.

**Writing – original draft:** Hansani Niroshika, Harshi Weerakoon.

**Writing – review & editing:** Harshi Weerakoon, Nalaka Herath, Kosala Weerakoon, Prasad Jayasooriya.

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
