## [Decision Letter · Decision Letter 0]

15 Dec 2025

Dear Dr. Niroshika,

Thank you for submitting your manuscript to PLOS ONE. After careful consideration, we feel that it has merit but does not fully meet PLOS ONE’s publication criteria as it currently stands. Therefore, we invite you to submit a revised version of the manuscript that addresses the points raised during the review process.

We look forward to receiving your revised manuscript.

Kind regards,

Jie Yang, M.D.

Guest Editor

PLOS One

Journal Requirements:

4. Please note that funding information should not appear in any section or other areas of your manuscript. We will only publish funding information present in the Funding Statement section of the online submission form. Please remove any funding-related text from the manuscript.

Reviewer's Responses to Questions

**Comments to the Author**

1. Does the manuscript provide a valid rationale for the proposed study, with clearly identified and justified research questions?

Reviewer #1: Yes

Reviewer #2: Yes

Reviewer #3: Yes

2. Is the protocol technically sound and planned in a manner that will lead to a meaningful outcome and allow testing the stated hypotheses?

Reviewer #1: Yes

Reviewer #2: Partly

Reviewer #3: Yes

3. Is the methodology feasible and described in sufficient detail to allow the work to be replicable?

Reviewer #1: Yes

Reviewer #2: Yes

Reviewer #3: Yes

4. Have the authors described where all data underlying the findings will be made available when the study is complete?

The PLOS Data policy requires authors to make all data underlying the findings described in their manuscript fully available without restriction, with rare exception, at the time of publication. The data should be provided as part of the manuscript or its supporting information, or deposited to a public repository. For example, in addition to summary statistics, the data points behind means, medians and variance measures should be available. If there are restrictions on publicly sharing data—e.g. participant privacy or use of data from a third party—those must be specified.requires authors to make all data underlying the findings described in their manuscript fully available without restriction, with rare exception, at the time of publication. The data should be provided as part of the manuscript or its supporting information, or deposited to a public repository. For example, in addition to summary statistics, the data points behind means, medians and variance measures should be available. If there are restrictions on publicly sharing data—e.g. participant privacy or use of data from a third party—those must be specified.

Reviewer #1: Yes

Reviewer #2: Yes

Reviewer #3: Yes

5. Is the manuscript presented in an intelligible fashion and written in standard English?

Reviewer #1: Yes

Reviewer #2: Yes

Reviewer #3: Yes

You may also provide optional suggestions and comments to authors that they might find helpful in planning their study.

Reviewer #1: Thank you for raising this search to identify potential new culprits implicated in CKD progression an identify gaps in knowledge.

Reviewer #2: This protocol outlines a scoping review based on the Arksey & O’Malley and JBI frameworks to systematically map the evidence on the role of TRAIL (TNFSF10) and its receptors (DR4, DR5, DcR1, DcR2, OPG) in chronic kidney disease (CKD). The authors plan to include human, animal, and in vitro studies, focusing on TRAIL expression, regulatory mechanisms, and associations with disease progression and clinical outcomes.

If executed rigorously, this scoping review can serve as a critical foundation for future prognostic or interventional studies, making its clinical potential meaningful.

[Major Concerns (Required Revisions)]

/Database coverage is incomplete.

EMBASE and Web of Science should be added, as they capture much of the biomedical and pharmacological literature.

Clinical trial registries (ClinicalTrials.gov, WHO ICTRP) and preprint servers (medRxiv, bioRxiv) should also be considered.

/Incomplete search strategy reporting.

The manuscript provides only partial search terms.

A full Boolean search string for each database (including synonyms such as TNFSF10, APO2L, CD253) should be added as a Supplementary File.

/Language bias.

Restricting to English-only publications may exclude relevant non-English epidemiologic or regional studies. Consider including papers with English abstracts or at least acknowledging this limitation explicitly.

/Clarify the purpose of MMAT quality assessment.

Scoping reviews typically do not exclude studies based on quality. Please clarify whether MMAT will be used descriptively (to summarize evidence quality) rather than for exclusion.

/Timeline inconsistency.

The text states “articles published up to July 2025” but does not specify the actual search date.Indicate the exact date of the search (e.g., Search conducted on August 31, 2025).

/Outcome definition unclear.

“Clinical outcomes” should be better defined (e.g., mortality, dialysis initiation, eGFR decline rate, cardiovascular events). Separate primary and secondary outcomes where possible.

/Assay variability not addressed.

Measurement of soluble TRAIL is highly method-dependent (different ELISA kits, storage conditions).Include assay type and sample source (plasma vs tissue) as data extraction items.

This is a well-structured and potentially impactful protocol.

By improving transparency in your search and data extraction methods, and by clearly defining how findings could inform biomarker or therapeutic development, your review will provide genuine value to both nephrology and translational immunology communities.

Reviewer #3: The manuscript is well-structured, clearly written, and methodologically sound. The authors have followed established frameworks (Arksey & O’Malley, JBI) and adhered to reporting guidelines (PRISMA-ScR), and the protocol is appropriately registered. The topic is relevant and addresses a notable gap in synthesizing the role of TRAIL in CKD.

Key Questions and Comments:

1. The PCC framework includes “all human studies, animal models, cell lines, and computational models.” Given the breadth of study types, how will the authors ensure a meaningful synthesis across such heterogeneous data? Will separate analyses or subgroup summaries be provided?

2. Table 2 is detailed and appropriate. However, it may be beneficial to add a field for “type of CKD stage” (e.g., CKD stages 1-5, ESRD) under “Population characteristics” to facilitate staging-related analysis. Additionally, the authors mention thematic analysis for qualitative data. Will a specific framework (e.g., Braun & Clarke) be used? If so, please specify.

3. The mixed-method appraisal tool (MMAT 2018) has been selected for quality appraisal, but the protocol states that “no studies will be excluded solely based on low scores.” How will low-quality studies (e.g., those with an MMAT score <50%) be handled in the synthesis? Will they be flagged, or will their findings be weighted differently to avoid misleading conclusions?

4. In the Introduction, the authors state: “Chronic kidney disease (CKD) is defined as kidney damage or glomerular filtration rate (GFR) <60 mL/min/1.73 m² for 3 months or more… affecting more than 700 million people worldwide.” The KDIGO 2021 update—which revises CKD staging and prevalence estimates—has not been cited. This outdated definition weakens the background. The protocol should reference the latest KDIGO guidelines (e.g., 2021 or 2024) for the definition of CKD.

5. Please include a concept summary figure or graphical abstract to explain the entire implementation plan.

Minor Corrections & Suggestions:

• Abstract line 37: Rephrase “Google Scholar reference lists of review articles as grey literature sources” for clarity. Suggested revision: “Reference lists of relevant reviews retrieved from Google Scholar will also be screened for grey literature.”

• Line 47-48: The definition of CKD is from 2005. Please consider citing more recent KDIGO guidelines (e.g., the 2024 update).

• Line 183-190 (Synthesis methods): Consider clarifying whether a meta-analysis is possible or if a narrative synthesis alone will be used.

.

Reviewer #1: No

Reviewer #2: No

Reviewer #3: No

---

## [Author Response · Author response to Decision Letter 1]

28 Jan 2026

Journal requirements

1 Please ensure that your manuscript meets PLOS ONE's style requirements, including those for file naming. The PLOS ONE style templates can be found at

Revised the manuscript according to the PLOS one journal style based on the given sample file.

2 Please note that funding information should not appear in any section or other areas of your manuscript. We will only publish funding information present in the Funding Statement section of the online submission form. Please remove any funding-related text from the manuscript

All funding related information has been removed from the manuscript text and is only available on the online submission form.

3 When completing the data availability statement of the submission form, you indicated that you will make your data available on acceptance. We strongly recommend all authors decide on a data sharing plan before acceptance, as the process can be lengthy and hold up publication timelines. Please note that, though access restrictions are acceptable now, your entire data will need to be made freely accessible if your manuscript is accepted for publication. This policy applies to all data except where public deposition would breach compliance with the protocol approved by your research ethics board. If you are unable to adhere to our open data policy, please kindly revise your statement to explain your reasoning and we will seek the editor's input on an exemption. Please be assured that, once you have provided your new statement, the assessment of your exemption will not hold up the peer review process

No datasets were generated or analysed during this scoping review protocol. All information required for the protocol is included in the manuscript and its supplementary materials. Data generated during the conduct of the scoping review will be made publicly available upon completion of the study. -

4 Please note that funding information should not appear in any section or other areas of your manuscript. We will only publish funding information present in the Funding Statement section of the online submission form. Please remove any funding-related text from the manuscript.

All funding related information has been removed from the manuscript text and only available on the online submission form.

5 If the reviewer comments include a recommendation to cite specific previously published works, please review and evaluate these publications to determine whether they are relevant and should be cited. There is no requirement to cite these works unless the editor has indicated otherwise.

All recommended references were reviewed for relevance and cited.

Reviewer 1

Thank you for raising this search to identify potential new culprits implicated in CKD progression and identify gaps in knowledge. We thank the reviewer for the positive and encouraging feedback.

Reviewer 2

/Database coverage is incomplete.

EMBASE and Web of Science should be added, as they capture much of the biomedical and pharmacological literature.

Clinical trial registries (ClinicalTrials.gov, WHO ICTRP) and preprint servers (medRxiv, bioRxiv) should also be considered.

EMBASE and Web of Science were added to the electronic database search alongside PubMed/ MEDLINE, ScienceDirect, Scopus, CINAHL, and the Cochrane Library. Clinical trial registries (ClinicalTrials.gov and WHO ICTRP) will also be searched. Preprint servers are not included, as only peer-reviewed literature is considered.

/Incomplete search strategy reporting.

The manuscript provides only partial search terms.

A full Boolean search string for each database (including synonyms such as TNFSF10, APO2L, CD253) should be added as a Supplementary File.

A complete and reproducible Boolean search strategy for each database has been provided as supplementary file 1.

/Language bias.

Restricting to English-only publications may exclude relevant non-English epidemiologic or regional studies. Consider including papers with English abstracts or at least acknowledging this limitation explicitly.

Considered including all studies without language bias and translating into English where needed.

/Clarify the purpose of MMAT quality assessment.

Scoping reviews typically do not exclude studies based on quality. Please clarify whether MMAT will be used descriptively (to summarize evidence quality) rather than for exclusion.

Clarified that MMAT will be used descriptively to summarize methodological quality. No studies will be excluded based on the MMAT scores. Lower quality studies (MMAT <50%) will be flagged and interpreted cautiously.

/Timeline inconsistency.

The text states “articles published up to July 2025” but does not specify the actual search date. Indicate the exact date of the search (e.g., Search conducted on August 31, 2025).

Articles up to January 2026 will be included according to the current timeline. Indicated the exact date for article search as 1st March 2026.

/Outcome definition unclear.

“Clinical outcomes” should be better defined (e.g., mortality, dialysis initiation, eGFR decline rate, cardiovascular events). Separate primary and secondary outcomes where possible.

Included mortality, dialysis initiation and cardiovascular events

/Assay variability not addressed.

Measurement of soluble TRAIL is highly method-dependent (different ELISA kits, storage conditions). Include assay type and sample source (plasma vs tissue) as data extraction items.

Sample source has been already included, and data extraction has been expanded to include assay type in the section of TRAIL expression.

Reviewer 3

/The PCC framework includes “all human studies, animal models, cell lines, and computational models.” Given the breadth of study types, how will the authors ensure a meaningful synthesis across such heterogeneous data? Will separate analyses or subgroup summaries be provided?

We clarified that data from heterogenous populations (human studies, animal models, cell lines, and related computational models) will be synthesized separately proving subgroup summaries.

/Table 2 is detailed and appropriate. However, it may be beneficial to add a field for “type of CKD stage” (e.g., CKD stages 1-5, ESRD) under “Population characteristics” to facilitate staging-related analysis. Additionally, the authors mention thematic analysis for qualitative data. Will a specific framework (e.g., Braun & Clarke) be used? If so, please specify.

CKD stage has been added as a data extraction variable under population characteristics.

Qualitative findings will be analyzed using a descriptive approach, focusing on summarizing and mapping key characteristics and concepts reported in the included studies in line with the objectives of a scoping review. Therefore no specific thematic analysis will be performed.

/The mixed-method appraisal tool (MMAT 2018) has been selected for quality appraisal, but the protocol states that “no studies will be excluded solely based on low scores.” How will low-quality studies (e.g., those with an MMAT score <50%) be handled in the synthesis? Will they be flagged, or will their findings be weighted differently to avoid misleading conclusions?

Clarified that MMAT will be used descriptively to summarize methodological quality. No studies will be excluded based on the MMAT scores. Lower quality studies (MMAT <50%) will be flagged and interpreted cautiously.

/In the Introduction, the authors state: “Chronic kidney disease (CKD) is defined as kidney damage or glomerular filtration rate (GFR) <60 mL/min/1.73 m² for 3 months or more… affecting more than 700 million people worldwide.” The KDIGO 2021 update—which revises CKD staging and prevalence estimates—has not been cited. This outdated definition weakens the background. The protocol should reference the latest KDIGO guidelines (e.g., 2021 or 2024) for the definition of CKD.

Updated with KDIGO report 2024 as “Chronic kidney disease (CKD) is defined as “abnormalities of kidney structure or function, present for a minimum of 3 months, with implications for health” (1) is a global health burden affecting more than 800 million people worldwide (2).”

/Please include a concept summary figure or graphical abstract to explain the entire implementation plan.

A conceptual summary figure illustrating the scope, methodology, and synthesis plan of the scoping review has been added.

/Abstract line 37: Rephrase “Google Scholar reference lists of review articles as grey literature sources” for clarity. Suggested revision: “Reference lists of relevant reviews retrieved from Google Scholar will also be screened for grey literature.”

Revised as suggested

/Line 47-48: The definition of CKD is from 2005. Please consider citing more recent KDIGO guidelines (e.g., the 2024 update).

Most recent KDIGO guidelines were cited

/Line 183-190 (Synthesis methods): Consider clarifying whether a meta-analysis is possible or if a narrative synthesis alone will be used

Consistent with the objectives of this scoping review, a meta-analysis will not be conducted. Results will be presented as narrative descriptions while including counts, frequencies, and proportions arise from quantitative data as appropriate.

---

## [Decision Letter · Decision Letter 1]

2 Mar 2026

Mapping the role of tumor necrosis factor-related apoptosis-inducing ligand (TRAIL) and its receptors in chronic kidney disease- a scoping review protocol

PONE-D-25-42046R1

Dear Dr. Niroshika,

We’re pleased to inform you that your manuscript has been judged scientifically suitable for publication and will be formally accepted for publication once it meets all outstanding technical requirements.

Kind regards,

Jie Yang, M.D.

Guest Editor

PLOS One

Additional Editor Comments (optional):

Thanks for the authors' efforts to comprehensively improve your manuscript according to editor's and reviewers' comments. I am pleased to inform you that your paper can be accepted for publication now.

Reviewers' comments:

Reviewer's Responses to Questions

**Comments to the Author**

1. Does the manuscript provide a valid rationale for the proposed study, with clearly identified and justified research questions?

Reviewer #2: Yes

Reviewer #3: Yes

2. Is the protocol technically sound and planned in a manner that will lead to a meaningful outcome and allow testing the stated hypotheses?

Reviewer #2: Yes

Reviewer #3: Yes

3. Is the methodology feasible and described in sufficient detail to allow the work to be replicable?

Reviewer #2: Yes

Reviewer #3: Yes

4. Have the authors described where all data underlying the findings will be made available when the study is complete?

The PLOS Data policy requires authors to make all data underlying the findings described in their manuscript fully available without restriction, with rare exception, at the time of publication. The data should be provided as part of the manuscript or its supporting information, or deposited to a public repository. For example, in addition to summary statistics, the data points behind means, medians and variance measures should be available. If there are restrictions on publicly sharing data—e.g. participant privacy or use of data from a third party—those must be specified.requires authors to make all data underlying the findings described in their manuscript fully available without restriction, with rare exception, at the time of publication. The data should be provided as part of the manuscript or its supporting information, or deposited to a public repository. For example, in addition to summary statistics, the data points behind means, medians and variance measures should be available. If there are restrictions on publicly sharing data—e.g. participant privacy or use of data from a third party—those must be specified.

Reviewer #2: Yes

Reviewer #3: Yes

5. Is the manuscript presented in an intelligible fashion and written in standard English?

Reviewer #2: Yes

Reviewer #3: Yes

You may also provide optional suggestions and comments to authors that they might find helpful in planning their study.

Reviewer #2: The authors have addressed all of my previous concerns thoroughly and sincerely. The additional data provided have significantly strengthened the manuscript. I am now satisfied with the quality of the work and believe it meets the standards for publication.

Reviewer #3: After revision, this manuscript meets the requirements for publication in the journal and is approved for publication.

.

Reviewer #2: No

Reviewer #3: No

---

## [Editor Report · Acceptance letter]

PONE-D-25-42046R1

PLOS One

Dear Dr. Niroshika,

I'm pleased to inform you that your manuscript has been deemed suitable for publication in PLOS One. Congratulations! Your manuscript is now being handed over to our production team.

Kind regards,

on behalf of

Dr. Jie Yang

Guest Editor

PLOS One